# In Situ Recrystallization of Mesoporous Carbon–Silica Composite for the Synthesis of Hierarchically Porous Zeolites

**DOI:** 10.3390/ma13071640

**Published:** 2020-04-02

**Authors:** Jun Du, Yan Wang, Yan Wang, Ruifeng Li

**Affiliations:** 1College of Chemistry and Chemical Engineering, Taiyuan University of Technology, Taiyuan 030024, China; jundu_2003@163.com (J.D.); rfli@tyut.edu.cn (R.L.); 2School of Chemistry and Chemical Engineering, Shanxi Datong University, Datong 037009, China; wangyan0355624@126.com

**Keywords:** hierarchical zeolites, in situ, synthesis, porous materials

## Abstract

Hierarchically porous ZSM-5 was prepared by utilizing a two-step crystallization procedure with carbon–silica composites as precursors. The hierarchically porous zeolites obtained a regular mesoporous structure with aluminum incorporated into the carbon–silica composite frameworks. The carbon–silica composite zeolites were characterized by XRD, TEM, SEM, and nitrogen adsorption/desorption. As-prepared hierarchical zeolites were used in the 1,3,5-triisopropylbenzene (TIPB) cracking reaction and exhibited significantly high TIPB conversion, while the accessibility factors were also determined.

## 1. Introduction

Zeolites are crystallized aluminosilicates with uniform micropores (*d*_micro_ < 2 nm) which have been widely utilized for ion exchange, catalysis, and adsorption [1,2,3]. However, conventional zeolites often lead to diffusion and mass transfer obstacles due to the nature of micropores, resulting in a relatively low utilization of the zeolite active sites in catalysis. Meanwhile, mesoporous materials (2 nm < *d*_meso_ < 50 nm) with amorphous pores have not been able to replace zeolites as catalysts in practical applications, owing to their low hydrothermal and thermal stability as well as their weak acidity [4,5,6,7]. In order to overcome these limitations, the synthesis of novel materials which combine both zeolites and ordered mesoporous materials is essential, especially for the field of catalysis. To date, the synthetic methods mainly utilize two approaches, such as bottom-up means with templates (porous carbons, cationic polymers or surfactants, etc.) and top-down means for the extraction of alumina or silica [8,9,10,11,12]. As-made micro-meso-large-pore zeolites have some important advantages, such as their hierarchical pore system, their increased number of active sites, as well as their better diffusivity and hydrothermal and thermal stabilities. However, low structural stability as well as the synthesis of materials with poor pore channel ordering and wide ranges of pore size distribution are still considerable challenges.

The recrystallization or transformation of carbon–silica composites and hollow silica gel into ZSM-5 has been reported by using tetrapropylammonium hydroxide (TPAOH) as a template [13,14]. Chen et al. [15] reported that hierarchical MCM-41/MFI composites prepared by the ion exchange of MCM-41 with tetrapropylammonium bromide and a subsequent steam-assisted recrystallization procedure, while transforming amorphous silica into zeolite crystal and composite inevitably led to the MCM-41 framework collapsing and a phase separation problem during the zeolite and mesoporous phases. In order to avoid the rapid collapse of the framework, Zhao et al. [13] reported that hierarchical ZSM-5 can be synthesized via the “in situ” transformation of mesoporous carbon–silica composite. In the first stage, a two-stage method was used to form the ordered mesoporous carbon–silica composite through a “tri-constituent” co-assembly of resin, silica, and surfactant. In the second stage, the water-vapor-assisted transformation was enforced in the mesoporous channels of carbon–silica composites infused with TPAOH. The mesostructure of carbon–silica composite was well retained, but the formation of ZSM-5 crystallites was restrained.

Herein, the hierarchical ZSM-5 zeolites were obtained through a two-step crystallization procedure via the carbon–silica composites being used as precursors. Firstly, highly ordered mesoporous carbon–silica composites were successfully synthesized by use of a hydrothermal synthetic method using furfuryl alcohol (FA) as organic carbon source and triblock copolymer P123 as the mesoporous template. The carbon–silica composites were then transformed into hierarchical ZSM-5 zeolites through steam-assisted dry-gel conversion.

## 2. Materials and Methods

### 2.1. Material Preparation

Typically, 2.0 g EO_20_PO_70_EO_20_ (P123, M = 5800, Aldrich Chemical Co., St. Louis, MS, USA) was entirely dissolved in HCl solution (60 mL, 2.0 M), then 3.3 g acetic acid was added (HAc, AR, Aldrich) into the solution. Then, 1.66 g of furfuryl alcohol (FA, AR, Aldrich) and 4.16 g of tetraethylorthosilicate (TEOS, 98%, AR, Aldrich) was added dropwise into the solution and stirred at 311 K for 24 h. The obtained precipitates were filtered and washed 3 times. The precipitates were mixed with 40 g of DI water and the pH was adjusted to 2 using a 2 M HCl solution. Finally, the obtained mixtures were transferred into a Teflon-lined autoclave at 423 K for 24 h. The final products were filtered, washed, dried at room temperature, and then carbonized under nitrogen atmosphere in a furnace set to 573 K at a heating rate of 1 K/min. Then, the temperature was increased to 1023 K at 5 K/min and maintained for 6 h. Next, the solid sample was obtained, which was designated as MCSI-0.4, where 0.4 stands for the mass ratio between FA and TEOS. For comparison, the Al-SBA-15 sample was synthesized with the same procedure, except that no organic matter was added to the gel system.

The hierarchically porous ZSM-5 was synthesized by incorporating alumina into MCSI-0.4 via steam-assisted in situ transformation. First, 0.08 g sodium aluminates (AR, Aldrich) was dissolved in 5.0 g of tetrapropylammonium hydroxide solution (TPAOH, 25%, Aldrich). Then, 1 g MCSI-0.4 composite was sequentially added and stirred at room temperature for 6 h. The final mixtures were recrystallized at 413 K for 7–12 h in an autoclave with enough water to produce saturated vapor. The resulting solid was washed with DI water and dried at 373 K overnight. Finally, the powder was calcined in N_2_ to 823 K at 1 K/min, then switched to air and calcined at 823 K for 5 h. The resultant hierarchically porous ZSM-5 material is referred to as HPZ-CSI-*x*, with *x* representing the crystallization time of in situ recrystallization. Traditional microporous ZSM-5 (Si/Al = 29) was prepared from the mixture with the molar composition: Al_2_O_3_:60SiO_2_:21.5TPAOH:650H_2_O [16].

Then, the prepared hierarchically porous ZSM-5 samples were further hydrothermally treated at 373 K for 7 d. The obtained samples were ion exchanged in 0.5 mol·L^−1^ NH_4_NO_3_ solutions at 353 K, with the samples then washed and calcined at 823 K to obtain the HPZ-CSI-*x* zeolites. Meanwhile, HPZ-CSI-8h sample has been treated in boiling water for 7 days to investigate the hydrothermal stability and denoted as HPZ-CSI-8H.

### 2.2. Material Characterizations

XRD was performed with a Shimadzu XRD-6000 instrument (Shimadzu Ltd., Kyoto, Japan) with Cu Kα radiation at 40 kV. N_2_ adsorption/desorption isotherms were measured at 77 K on Quantachrome NOVA 1200e (Quantachrome Instruments, Boynton Beach, FL, USA). The sample was degassed in vacuum at 613 K for 5 h. The specific surface areas (*S*_BET_) were calculated via the Brunauer–Emmett–Teller (BET) method, using adsorption branch in pressure ranging from 0.05 to 0.15. Total pore volume *V*_t_ was estimated from the adsorbed amount at *P/P_0_* = 0.99. Pore size distribution (PSD) was obtained by applying the adsorption model of nonlocal density functional theory (NLDFT method). Scanning electronic microscopy (SEM) images were obtained with a HITACHI S-4800 at 1 kV (HITACHI Ltd., Tokyo, Japan). Transmission electron microscopy (TEM) images were obtained with a JEOL 2100 microscope at 200 kV (JEOL Ltd., Tokyo, Japan).

### 2.3. Catalytic Tests

The catalytic experiments were conducted in a micro-reactor unit (shown in Appendix A). The catalyst (sized 0.8–1.0 mm) was firstly treated at 773 K for 5 h in air. Three cracking reactions with different catalyst-to-reactant ratios (1:1, 1:2, 1:3) of 1,3,5-triisopropylbenzene (TIPB) were conducted at 773 K and for 60 s with 200 mg of catalyst, with a purging nitrogen flow of 20 mL/min. A gas chromatograph (GC, Agilent 7890B, Agilent Technologies Co. Ltd., Santa Clara, CA, USA) equipped with a flame ionization detector (FID) and D2887 column was used to detect the products.

By fitting the conversions to Equation (1), the rate constants (*k*) were calculated. The volumetric expansion factor is listed as shown in Equation (2).
*k* = −(cat ∙ reactant^−1^ ∙ TOS)^−1^ ∙ [*ε*X + (1 + *ε*)ln(1−X)](1)
*ε* = (Σ molar selectivity of products)^−1^(2)

## 3. Results and Discussion

SEM images were utilized to investigate the morphology of SBA-15, MCSI-0.4, and HPZ-CSI-*x* materials. As shown in Figure 1a, SBA-15 obtained a uniform particle size, which indicates that the template was prepared well. MCSI-0.4 obtained the same morphology as SBA-15, which suggests that the hydrothermal treatment could form a stable carbon–silica composite. Figure 1d,e,f shows the fine structure of HPZ-CSI-*x*. HPZ-CSI-*x* showed a honeycomb-like ball-shaped morphology with an average diameter of ca. 1–2 μm. The EDS analysis results showed that the Si/Al of HPZ-CSI-*x* were almost the same as with the synthesis gel (Si/Al = 20, Appendix A). SEM results proved that the HPZ-CSI-*x* showed a nanocrystals assemblage morphology, suggesting that the presence of carbon materials resulted in the confinement of the crystal growth of zeolite. This was further confirmed by the pore size distribution and XRD. After the prepared hierarchically porous ZSM-5 samples were further hydrothermally treated at 373 K for 7 d, the HPZ-CSI-*x* showed the morphology of sphere particles composed of nanocrystal, which appeared rather dense and smooth (Figure 1e,f).

As revealed by the corresponding HRTEM images of MCSI-0.4 and HPZ-CSI-8h materials, MCSI-0.4 clearly showed the lattice fringes of the silica–carbon composites with about 4 nm thickness of the mesopore wall and about 5–6 nm mesopore size (Figure 2a). HPZ-CSI-8h also showed that the lattice fringes were implanted in the mesoporous framework (Figure 2b,c). The particle size of HPZ-CSI-8h nanocrystals at about 10 nm are well matched with the unit cell parameters of the MCSI-0.4 precursors. On closer observation, the HRTEM image in Figure 2c reveals the crystalline structure of HPZ-CSI-8h, which confirms the mesoporous walls were zeolitized in situ. The lattice fringe of the HPZ-CSI-8h had a dot-shaped distribution, which is in accordance with the SEM results. An intercrystalline pore channel of about 10–20 nm can be clearly observed. Mesostructured HPZ-CSI-*x* with spherical nanocrystal particle morphology was obtained. The interspace between the nanocrystals particles could be visualized, suggesting that the removal of carbon materials resulted in the generation of a mesoporous structure, which was further confirmed by TG/DTA analysis. The weight loss of the MCSI-0.4 samples in TG analysis was 20.1% (Appendix A). The presence of carbon materials should be conducive to the combustion of a structure-directing agent. The intercrystalline gaps between the nanocrystals were randomly distributed.

The structures of the synthesized SBA-15, MCSI-0.4, and HPZ-CSI-*x* samples were investigated by low-angle XRD. The resulting XRD pattern for SBA-15, obtained in the 2*θ* range of 0.6°–7°, is presented in Figure 3a, together with those of the MCSI-0.4 and HPZ-CSI-*x* samples for comparison. The low-angle XRD pattern for the SBA-15 sample presented three distinct peaks attributed to (100), (110), and (200) lattice planes, which were characteristics of meso-structures belonging to a 2D hexagonal P6mm space group. Compared with SBA-15, MCSI-0.4 samples also showed meso-structures with P6mm symmetry, as seen in the presence of (100), (110), and (200) diffraction peaks, suggesting that the organic carbon source (furfuryl alcohol) was involved in binding the framework of mesoporous materials during the formation of the mesostructured silica–carbon composites. As shown in Figure 3a, the diffraction peaks of MCSI-0.4 were slightly shifted to higher values, which can be attributed to structural shrinkage during carbonization at high temperature (1023 K) [17,18]. Weaker (100), (110), and (200) diffraction peaks of MCSI-0.4 samples, compared to SBA-15, were the result of introducing an organic carbon source, which caused the structural regularity of the meso-structured silica-carbon composites to decrease. However, for HPZ-CSI-x, there was only one weak characteristic peak at 2*θ* = 1° Figure 3b, and the diffraction peaks of (110) and (200) crystal planes almost disappeared. The intensity of the diffraction peaks reflects the long-range order of the materials. After being transformed in situ with the assistance of water vapor, the intensity of the diffraction peaks ascribed to (110) and (200) crystal planes nearly disappeared. This is due to the recrystallization of MCSI-0.4 silica into ZSM-5 type zeolite crystal, while there was also a heavy structure distortion on the precursors. Accordingly, the long-range ordered structure of the synthesized materials gradually became more disordered.

The wide-angle XRD patterns of HPZ-CSI-*x* samples are shown in Figure 3c. All materials exhibited a highly crystalline MFI topology structure, while no other impurity phase could be observed when the recrystallization time was greater than 3 h. In contrast to conventional ZSM-5, the HPZ-CSI-*x* samples exhibited broader diffraction peaks, indicating the presence of small domains. By comparison, HPZ-CSI-3h exhibited relatively low crystallinity. Meanwhile, HPZ-CSI-8h showed a notable increase of crystallinity, which indicates that the crystallization of the amorphous silica wall significantly accelerated with the evolution of the crystallization time.

N_2_ adsorption/desorption isotherms and NLDFT pore size distribution of the materials are presented in Figure 4 and Appendix A. All samples exhibited type IV isotherms with an H1 hysteresis loop, suggesting the presence of mesopores. An obvious hysteresis loop at a relative pressure of 0.4–0.8 was observed in all the isotherms, indicating that these materials had the characteristics of a concentrated distribution of mesopores. The isotherm of MCSI-0.4 showed a quite steep hysteresis loop which was similar to that of SBA-15, indicating the preparation of these precursor materials (Appendix A).

The structural parameters of the materials are summarized in Table 1. The pore size of the HPZ-CSI-*x* was ca. 6.1 nm. The total pore volume of HPZ-CSI-8h was 0.58 cm^3^/g, which consisted of 0.14 cm^3^/g microporous volume and 0.44 cm^3^/g mesoporous volume. The BET surface area of HPZ-CSI-8h was ca. 526 m^2^/g, consisting of 226 m^2^/g microporous area and 300 m^2^/g external surface area. The structural parameters of the HPZ-CSI-*x* were damaged severely with the evolution of the crystallization time. This may have been due to the recrystallization transformation of amorphous silica wall into ZSM-5-type zeolite nanocrystals during the in situ transformation, assisted with water vapor, while the flake of these zeolite nanocrystals coming off led to the mesopore structure being damaged. As a comparison, the total pore volume of HPZ-CSI-8H (treated in the boiling DI water for seven days) was 0.53 cm^3^/g, consisting of 0.15 cm^3^/g microporous volume and 0.38 cm^3^/g mesoporous volume. The BET surface area of HPZ-CSI-8H was ca. 445 m^2^/g, consisting of 255 m^2^/g microporous area and 190 m^2^/g external surface area. Before and after hydrothermal treatments, the external surface area and mesoporous volume of the HPZ-CSI-8h sample revealed a slight decrease, indicating the collapse of the residual no-crystallization amorphous silica wall. However, the micropore area and other pore structure parameters of the HPZ-SIC-8h sample were not obviously damaged during the treatment in boiling DI water for seven days. This result suggests that the material obtained a remarkable hydrothermal stability.

The hierarchy factor (HF) is defined as (*V*_micro_/*V*_total_) × (*S*_Ext_/*S*_BET_), which is an important indicator for classifying zeolites by their porosity [19]. The HF of ZSM-5 was only 0.07, while the HFs of HPZ-CSI-*x* (*x* = 3, 8) were 0.07 and 0.14, respectively, revealing that HPZ-CSI-*x* obtained a more widespread hierarchy system. According to the literature [19], the highest HF value of the hierarchical ZSM-5 was 0.15, which was obtained via different kinds of treatment methods, such as hard-template, soft-template, and post-synthesis.

Since vapor treatment was essential for in situ crystallization, the presence of extra mesopores in MCSI-0.4 presumably acted as both vapor reservoirs and mesoreactors for the crystallization of the amorphous silica. Therefore, the mesopores with uniform pore size were favorable for in situ transformation assisted processes [20].

The cracking reaction of bulky molecules is a heavy oil cracking reaction under the effect of a catalyst. During the catalytic cracking process, the bulky molecules are transformed into small-molecule gas, gasoline, light diesel oil, etc. Ogura [21] reported on the preparation of hierarchical ZSM-5 through alkali treatment, and showed better catalyst activity in the isopropyl benzene cracking reaction. Though the acidic properties changed little, the hierarchical zeolite with additional mesoporosity in the microporous structures could promote the diffusion properties of isopropyl benzene in zeolite intracrystalline. TOF values of the catalyst were double those of the untreated ZSM-5 zeolite. Jung [22] also reported on the preparation of hierarchical ZSM-5 by alkali treatment and showed a lower reaction conversion rate and higher catalytic selectivity in the octane cracking reaction. Christensen [23] reported a carbon template synthesis route of hierarchical ZSM-5. The *n*-hexadecane conversion rate was 12% higher than the traditional microporous ZSM-5 in *n*-hexadecane cracking reactions. As described above, the hierarchical zeolites obtained higher acidity and better accessibility of active sites, which can be attributed to the presence of the additional meso- or macroporosity. Higher activity has been clearly exhibited in reactions with large molecules. This could be ascribed to more accessible active sites and shorter diffusion paths compared with conventional zeolites. Thus, the accessible active sites were also evaluated on the basis of the bulky molecular cracking reaction. TIPB as a branched and bulky molecule (kinetic diameter = 0.95 nm) cannot pass through 10-MR channels. Thus, the cracking reaction of TIPB can reflect the changes of structural parameters and the accessibility of the active sites.

Cracking reactions of TIPB were conducted in order to investigate the impact of the structural and acidic properties on catalytic performance. The conversion at various reactant/catalyst ratios and kinetic constants was investigated. The conversion of TIPB for the hierarchical materials was higher than for the HZSM-5 zeolites. The hierarchical zeolites demonstrated an increased catalytic activity due to greater mesopore surface area and more active sites being exposed on the surface of the hierarchical zeolite (the acidity of HZSM-5 and HPZ-CSI-*x* was investigated by NH_3_-TPD and Py-FTIR (Appendix A)). However, HPZ-CSI-8h showed an outstanding medium-strong acidity which can be attributed to the confinement crystal growth of zeolite by carbon materials. Due to mostly acid sites existing in the microporous channels, as shown in Appendix A, the total acid amounts gradually increased with the evolution of the crystallization time. HPZ-CSI-8h obtained higher microporous areas and volumes in comparison to HPZ-CSI-3h and HPZ-CSI-5h. The pyridine adsorption FTIR results showed that the acidity of HPZ-CSI-8h was weaker than that of HZSM-5, which was in accordance with the NH_3_-TPD results. All of these provide more efficient reaction space and reaction active sites in the cracking reaction of TIPB due to the diffusion path shortening, which improves the diffusion in micropore channels and the accessibility of the active sites. Then, the tendency of undesirable hydrocarbon formation can be largely hindered by the introduction of meso/macro porosity into the interior of zeolite crystals. The results suggest that the hierarchical zeolites have excellent catalytic properties in TIPB cracking reactions involving bulky molecules. 

The conversion from the TIPB catalytic cracking at 623 K, 723 K, and 70 s time on stream over the HZSM-5 and HPZ-CSI-8h zeolites is displayed in Figure 5A. The activities of TIPB cracking on the hierarchical ZSM-5 are shown in Table 2. The hierarchical HPZ-CSI-8h showed higher activity, while the traditional microporous ZSM-5 zeolite produced lower conversion and kinetic constants. The TIPB conversion rate was higher (10%–20% at 723 K). The better catalytic performance for TIPB of hierarchical HPZ-CSI-8h samples was due to the higher mesoporous surface area and better accessibility of TIPB. This can be explained by considering that more acid active sites were exposed on the surface of the hierarchical zeolite. Therefore, the hierarchical zeolite showed superior acidity with high turnover frequency, which can be attributed to better accessibility of the active sites for TIPB. Finally, the generation of mesopores led to a relatively high TIPB conversion. As expected, despite possessing a lower acidity, the hierarchical HPZ-CSI-8h samples with large external surface areas (*V*_meso_/*V*_total_ = 0.72) and lower Si/Al molar ratios (Si/Al = 20) showed much higher benzene yields (wt %) for cracking reactions of TIPB, with a higher conversion of TIPB than HZSM-5 (*V*_meso_/*V*_total_ = 0.25, Si/Al = 29) with small external surface area (Figure 5B and Table 1). The micro/meso/macroporous characters of HPZ-CSI-8h allowed coke deposit formation at higher conversion than with HZSM-5 (Figure 5C).

The activity was also investigated via rate constants, which demonstrated the same trend as the accessibility and mesoporous surface area [24]. The reaction rate constants from TIPB cracking at 623 K, 723 K, and 70 s time on stream over the HZSM-5 and HPZ-CSI-8h samples are shown in Figure 6. The kinetic constants (*k_a_*) were obtained by fitting the conversions to the first-order kinetic equations, which were respectively 0.052 and 0.086 g_Reactant_g^−1^_Cat_s^−1^ for the HZSM-5 and HPZ-CSI-8h samples (723 K, 70 s time on stream and 0.33 catalyst/reactant ratio). The *k*_TIPB_ value of HPZ-CSI-8h was higher than those of the others. The catalytic activity increased along with the accessibility of the active sites, suggesting that the mesoporous structure can effectively promote the diffusion of TIPB. In conclusion, the acidity and accessibility of the active sites were the two most important factors affecting TIPB cracking performance. In other words, the superior cracking performance of the hierarchical zeolite can be attributed to the introduction of the interconnected mesopore structures, which provides more accessible active sites, and thus the molecular diffusion of bulky molecules is facilitated.

## 4. Conclusions

In this work, hierarchically porous ZSM-5 with unique mesoporous structures was prepared via a two-step crystallization procedure with carbon–silica composites as precursors. The hierarchically porous zeolites were demonstrated to not only have mesoporous regularity, but also the ability to successfully incorporate aluminum into the mesoporous carbon–silica composite framework through the steam-assisted dry-gel conversion method. The hierarchical structure and the aggregated nanocrystals of HPZ-CSI-*x* greatly improved the diffusion of bulky molecules. The HPZ-CSI-*x,* with high acidity and active site availability, could be employed in various organic transformations, especially due to its excellent catalytic performance (i.e., selectivity, activity, and lifetime) for bulky molecules.

## Figures and Tables

**Figure 1 materials-13-01640-f001:**
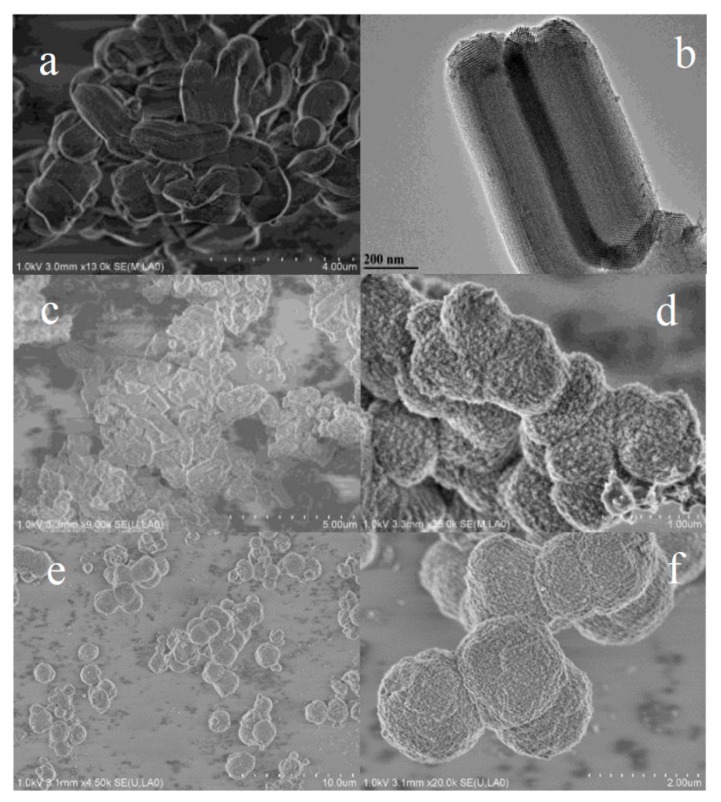
SEM images of SBA-15(**a**), MCSI-0.4 (**c**) (calcined at 1023K in N_2_), HPZ-CSI-*x* (**d**, **e**, **f**) (calcined at 823K in O_2_), (TEM images of SBA-15(**b**)).

**Figure 2 materials-13-01640-f002:**
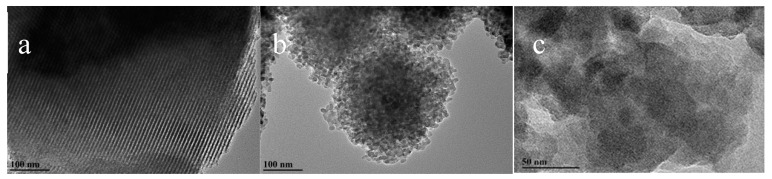
Images of (**a**) MCSI-0.4 calcined at 1023 K in N_2_; (**b**,**c**) HPZ-CSI-8h calcined at 823 K in O_2_.

**Figure 3 materials-13-01640-f003:**
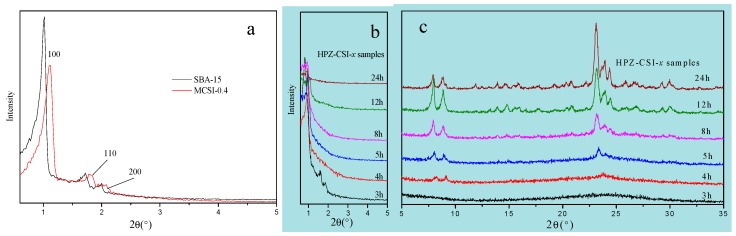
Small-angle (**a**,**b**) and wide-angle (**c**) powder XRD patterns of MCSI-0.4, SBA-15, and HPZ-CSI-x samples.

**Figure 4 materials-13-01640-f004:**
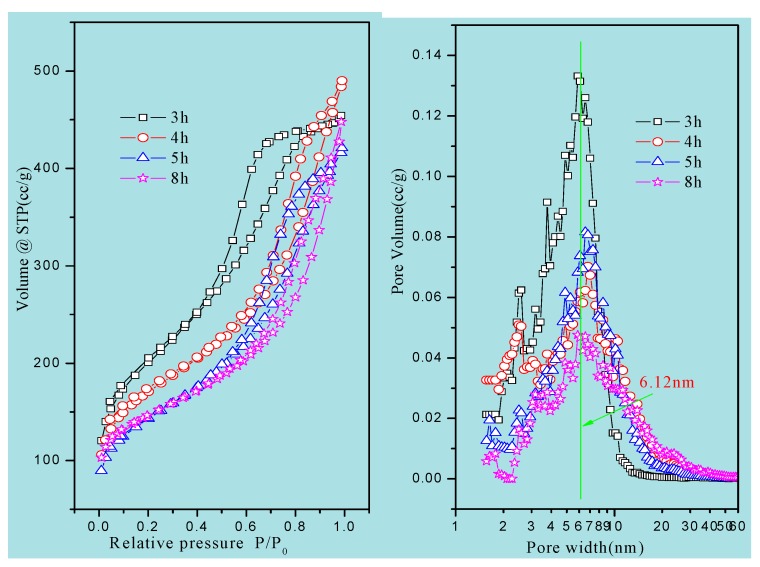
N_2_ adsorption/desorption isotherms and the corresponding pore size distributions of HPZ-CSI-*x* samples.

**Figure 5 materials-13-01640-f005:**
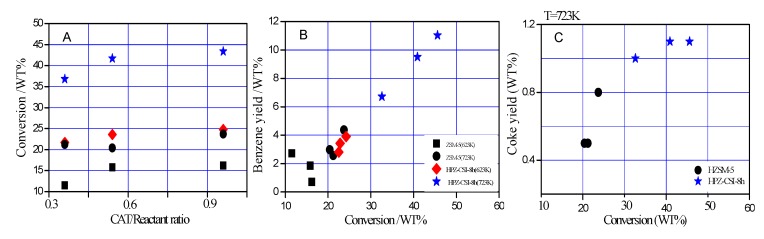
(**A**) Conversion and (**C**) coke yield of 1,3,5-triisopropylbenzene (TIPB) cracking reactions and (**B**) the benzene yields (wt %) for TIPB cracking reactions at 623 K, 723 K, and 70 s time on stream over the HZSM-5 (■ 623 K, ● 723 K) and HPZ-CSI-8h (◆ 623 K, ★ 723 K) zeolites.

**Figure 6 materials-13-01640-f006:**
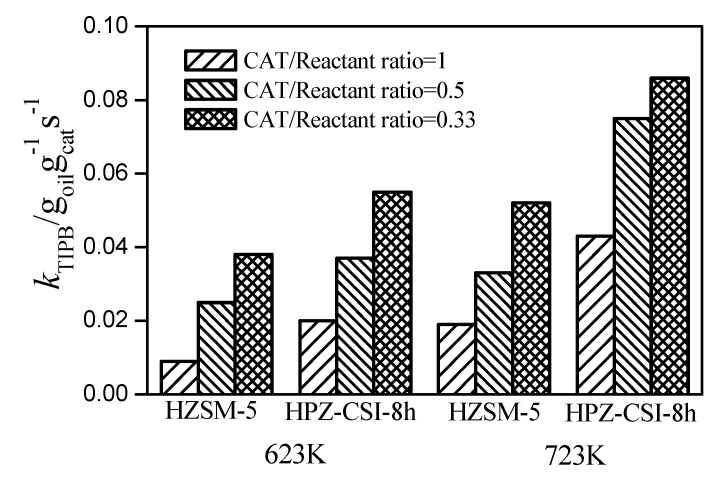
Rate constants (*k*_TIPB_) of TIPB cracking at 623 K, 723 K, and 70 s time on HZSM-5 and HPZ-CSI-8h.

**Table 1 materials-13-01640-t001:** Structural parameters and acidity of the samples. HF: hierarchy factor.

Sample	*S*_BET_ (m^2^/g)	*S*_micro_ (m^2^/g)	*S*_ext_ (m^2^/g)	*V*_micro_ (cm^3^g^−1^)	*V*_meso_ (cm^3^g^−1^)	*V*_meso_/*V*_total_	HF
SBA-15	654	33	621	0.024	1.04	--	0.02
ZSM-5	415	375	40	0.15	0.05	0.25	0.07
MCSI-0.4	598	45	553	0.022	0.48	--	0.04
HPZ-CSI-3h	616	139	477	0.06	0.62	--	0.07
HPZ-CSI-8h	526	226	300	0.14	0.44	--	0.14
HPZ-CSI-8H	445	255	190	0.15	0.38	0.72	0.12

**Table 2 materials-13-01640-t002:** Conversion, yield, selectivity, and rate constants (*k*_TIPB_) from the TIPB catalytic cracking at 623 K, 723 K, and 70 s time on stream over the HPZ-CSI-8h zeolites.

T (K)	Catalyst/Reactant Ratio (wt %)	Conversion of TIPB (wt %)	Yield (wt %)			Selectivity (wt %)			*k*_TIPB_ (g_reactant_g^−1^_cat_s^−1^)
			benzene	cumene	DIPB	benzene	cumene	DIPB	
**623**	**1**	**24.2**	3.90	0.80	19.50	16.10	3.32	80.58	0.020
	0.5	22.8	3.42	0.72	18.66	15.01	3.15	81.84	0.037
	0.33	22.5	2.81	0.60	19.09	12.51	2.63	84.86	0.055
723	1	45.6	11.03	1.59	32.98	24.18	3.48	72.34	0.043
	0.5	40.9	9.50	1.22	30.18	23.22	2.98	73.80	0.075
	0.33	32.6	6.72	0.83	25.05	20.62	2.55	76.83	0.086

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
