# Peer review of "In Situ Recrystallization of Mesoporous Carbon–Silica Composite for the Synthesis of Hierarchically Porous Zeolites"

_materials, 2020, doi:10.3390/ma13071640_

Round 1

Reviewer 1 Report

The authors investigated the in-situ recrystallization of mesoporous carbon-silica composites for synthesis  of hierarchically porous ZSM-5 zeolites.

This is an actual and interesting study and can be published in materials.

However, a major revison is necessary before publication.

The authors state that the hierarchically porous zeolites are characterized by a regular ordered mesopore structure,. However, this cannot be proven by the experimental results (TEM images 2b, c, XRD patterns 3b and N2 sorption isotherms Fig 4). The authors should show TEM images (like Fig 2a) from  which the regular mesopore structure in the hierarchical zeolites can been clearly seen or modify the manuscript.

Furthermore, the following things must be added:

chapter 2.1  synthesis of the pure SBA-15 and the reference HZSM-5 materials

chapter 3    nitrogen sorption isotherms of SBA-15, HZSM-5, MCSI-0.4, 

chapter 3    Si/Al molar ratio of all hierarchically structured HPZ-CSI-x zeolites and the reference zeolite HZSM-5 including a discussion in the context of the catalytic experiments (acidity).

Finally, the manuscript also contains many spelling mistakes.

Author Response

Dear editor,

We are grateful for the reviewers’ suggestions and comments concerning our manuscript (materials-750127). According these suggestions and comments, we carefully revised the manuscript and all the revisions were highlighted in red in the revised manuscript. The detailed changes and responses to the reviewers’ comments are listed as follows:

Reviewer 1:

Q1. The authors state that the hierarchically porous zeolites are characterized by a regular ordered mesopore structure,. However, this cannot be proven by the experimental results (TEM images 2b, c, XRD patterns 3b and N2 sorption isotherms Fig 4). The authors should show TEM images (like Fig 2a) from which the regular mesopore structure in the hierarchical zeolites can been clearly seen or modify the manuscript.

A1. The statement of the regular ordered mesopore structure is controversial. Thus, for accurate description, we modify the ordered mesopore structure into mesopore structure.

Q2. chapter 2.1  synthesis of the pure SBA-15 and the reference HZSM-5 materials

chapter 3    nitrogen sorption isotherms of SBA-15, HZSM-5, MCSI-0.4,

chapter 3    Si/Al molar ratio of all hierarchically structured HPZ-CSI-x zeolites and the reference zeolite HZSM-5 including a discussion in the context of the catalytic experiments (acidity).

A2. The information and data have been added in the supporting information and the revised manuscript.

Q3. Finally, the manuscript also contains many spelling mistakes.

A3. Spelling mistakes have been checked and revised.

Reviewer 2 Report

The manuscript presents the studies of o the synthesis of hierarchical porous zeolites based on recrystallization of amorphous mesoporous silica. The concept of the studies is very interesting and promising. In general the manuscript is well designed and prepared. I have only some questions and remarks to authors.

  1. More information about MAT (micro activity test) unit should be presented in the revised version of the manuscript. What type of detector(s) is used? Which reactants can be identified by this system? Producer?
  2. In the case of the catalyst for hydrocracking surface acidity of the catalyst is a very important issue. Have authors any information about surface concentration of acid sites? Is aluminum present in the zeolitic phase or maybe part of it formed a separate phases?
  3. Have authors any idea what is the stability of the studied catalysts during their regeneration (removal of coke)?

Author Response

Dear editor,

We are grateful for the reviewers’ suggestions and comments concerning our manuscript (materials-750127). According these suggestions and comments, we carefully revised the manuscript and all the revisions were highlighted in red in the revised manuscript. The detailed changes and responses to the reviewers’ comments are listed as follows:

Reviewer 2:

Q1. More information about MAT (micro activity test) unit should be presented in the revised version of the manuscript. What type of detector(s) is used? Which reactants can be identified by this system? Producer?

A1. The schematic diagram of micro activity test unit has been added in the Supporting Materials. And the detector and reactants information have been added in the revised manuscript.

Q2.  In the case of the catalyst for hydrocracking surface acidity of the catalyst is a very important issue. Have authors any information about surface concentration of acid sites? Is aluminum present in the zeolitic phase or maybe part of it formed a separate phases?

A2. Py-FTIR and NH3-TPD have been utilized to investigate the acidity of the catalysts, and the corresponding information have been added in the revised manuscript and Supporting Materials. The aluminum mostly exists in the zeolitic phase and no separate phase or extra framework aluminum phase have been observed in the catalysts.

Q3. Have authors any idea what is the stability of the studied catalysts during their regeneration (removal of coke)?

A3. The removal of coke has been analyzed by TG/DGA method. The result indicates that the coke can be removed and the mesoporous structure can be regenerated after calcination. TG/DGA data have added in the supporting materials and the discussion has been added in the revised manuscript.

Round 2

Reviewer 1 Report

The revised article can published in its present form.